# WEAKLY SUPERVISED KNOWLEDGE TRANSFER WITH PROBABILISTIC LOGICAL REASONING FOR OBJECT DETECTION

**Martijn Oldenhof**
ESAT - STADIUS
KU Leuven, Belgium
martijn.oldenhof@kuleuven.be

**Adam Arany**
ESAT - STADIUS
KU Leuven, Belgium
adam.arany@esat.kuleuven.be

**Yves Moreau**
ESAT - STADIUS
KU Leuven, Belgium
yves.moreau@esat.kuleuven.be

**Edward De Brouwer**
ESAT - STADIUS
KU Leuven, Belgium
edward.debrouwer@gmail.com

## ABSTRACT

Training object detection models usually requires instance-level annotations, such as the positions and labels of all objects present in each image. Such supervision is unfortunately not always available and, more often, only image-level information is provided, also known as weak supervision. Recent works have addressed this limitation by leveraging knowledge from a richly annotated domain. However, the scope of weak supervision supported by these approaches has been very restrictive, preventing them to use all available information. In this work, we propose ProbKT, a framework based on probabilistic logical reasoning that allows to train object detection models with arbitrary types of weak supervision. We empirically show on different datasets that using all available information is beneficial as our ProbKT leads to significant improvement on target domain and better generalization compared to existing baselines. We also showcase the ability of our approach to handle complex logic statements as supervision signal. Our code is available at https://github.com/molden/ProbKT

## 1 INTRODUCTION

Object detection is a fundamental ability of numerous high-level machine learning pipelines such as autonomous driving [4; 16], augmented reality [42] or image retrieval [17]. However, training state-of-the-art object detection models generally requires detailed image annotations such as the box-coordinates location and the labels of each object present in each image. If several large benchmark datasets with detailed annotations are available [26; 15], providing such detailed annotation on new specific datasets comes with a significant cost that is often not affordable for many applications.

More frequently, datasets come with only limited annotation, also referred to as *weak supervision*. This has sparked research in weakly-supervised object detection approaches [25; 6; 40], using techniques such as multiple instance learning [40] or variations of class activation maps [3]. However, these approaches have been shown to significantly underperform their fully-supervised counterparts in terms of robustness and accurate localization of the objects [39].

An appealing and intuitive approach to improve the performance of weakly supervised object detection is to perform transfer learning from an existing object detection model pre-trained on a fully annotated dataset [14; 46; 43]. This approach, also referred to as transfer learning or domain adaptation, consists in leveraging transferable knowledge from the pre-trained model (such as bounding boxes prediction capabilities) to the new weakly supervised domain. This transfer has been embodied in different ways in the literature. Examples include a simple fine-tuning of the classifier of bounding box proposals of the pre-trained model [43], or an iterative relabeling of the weakly supervised dataset for retraining a new full objects detection model on the re-labeled data [46].

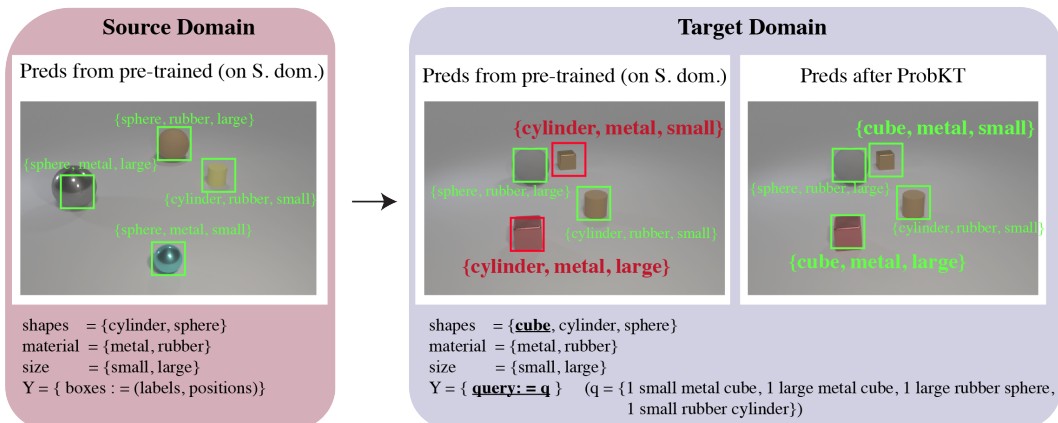

Figure 1: **ProbKT**: Weakly supervised knowledge transfer with probabilistic logical reasoning. (Left) A model can be trained on the source domain using full supervision (labels, positions) but only on a limited set of shapes (cylinders and spheres). (Middle) The pre-trained model does not recognize the cubes from the target domain correctly. (Right) The model can adapt to the target domain after applying ProbKT and can recognize the cubes.

However, existing approaches are very restrictive in the type of weak supervision they are able to harness. Indeed, some do not support new object classes in the new domain [20], others can only use a label indicating the presence of an object class [46]. However, in practice, the supervision on the new domain can come in very different forms. For instance, the count of each object class can be given, such as in atom detection from molecule images where only chemical formula might be given. Or, when many objects are present on an image, a range can be provided instead of an exact class counts (*e.g.* "there are *at least* 4 cats on this image"). Crucially, this variety of potential supervisory signals on the target domain cannot be fully utilized by existing domain adaption approaches.

To address this limitation, we introduce ProbKT, a novel framework that allows to generalize knowledge transfer in object detection to arbitrary types of weak supervision using neural probabilistic logical reasoning [27]. This paradigm allows to connect probabilistic outputs of neural networks with logical rules and to infer the resulting probability of particular queries. One can then evaluate the probability of a query such as "the image contains at least two animals" and differentiate through the probabilistic engine to train the underlying neural network. Our approach allows for arbitrarily complex logical statements and therefore supports weak supervision like class counts or ranges, among other. To our knowledge, this is the first approach to allow for such versatility in utilizing the available information on the new domain.

To assess the capabilities of this framework, we provide extensive empirical analysis of multiple object detection datasets. Our approach also supports any type of objects detection backbone architecture. We thus use two popular backbone architectures, DETR [7] and RCNN [34] and evaluate their performance in terms of accuracy, convergence as well of generalization on out-of-distribution data. Our experiments show that, due to its ability to use the complete supervisory signal, our approach outperforms previous works in a wide range of setups.

*Key contributions*: (1) We propose a novel knowledge transfer framework for object detection relying on probabilistic programming that uniquely allows using arbitrary types of weak supervision on the target domain. (2) We make our approach amenable to different levels of computational capabilities by proposing different approximations of ProbKT. (3) We provide an extensive experimental setup to study the capabilities of our framework for knowledge transfer and out-of-distribution generalization.

## 2 RELATED WORKS

A comparative summary of related works is given in Table 1. We distinguish three main categories: (1) pure weakly supervised object detection methods (WSOD) that do not leverage a richly annotated source domain, (2) unsupervised object detection methods with knowledge transfer (DA or domain adaptation methods) that do not use supervision on the target domain and (3) weakly supervised

object detection methods with knowledge transfer (WSOD w/transfer) that are restrictive in the type of supported weak supervision. To our knowledge, our work is the first to allow for arbitrary supervision on the target domain (and supporting new classes in the target domain) while also leveraging knowledge from richly annotated domains. ProbKT supports arbitrary weak supervision thanks to the inherited expressiveness of Prolog [41] which is based on a subset of first-order predicate logic, Horn clauses and is Turing-complete.

**Weakly supervised object detection (WSOD)** This class of method allows training object detection models with only weak supervision. One can thus train these approaches directly on the target domain. However, they do not allow to leverage potentially available richly annotated datasets, which has been shown to lead to worse performance [39]. Different flavors of WSOD architectures have been proposed relying on a variety of implementations such as multiple instance learning (MIL)-based [25; 40] or class activation (CAM) based [47; 3]. In contrast to WSOD methods, our approach is designed to exploit existing richly annotated datasets and thus provides increased performance on the target domain. For a comprehensive review of WSOD methods we refer the reader to Shao et al. [39].

**Domain adaptation methods (DA)** In contrast to WSOD methods, domain adaptation methods do rely on fully supervised source domain dataset. However, they do not assume any supervision on the target domain and are therefore not equipped to exploit such signal when available [37; 8; 22; 48].

**WSOD with knowledge transfer** Our approach belongs to the class of weakly supervised object detection models with knowledge transfer. These methods aim to transfer knowledge from a source domain, where full supervision is available, to a target domain where only weak labels are available. Existing work in this class of models only allows for limited type of supervision of the target domain. Most architectures only support a label indicating the presence or absence of a class of object in the image[14; 46; 43]. Inoue et al. [20] allows for class counts as weak supervision but unfortunately does not allow for new classes in the target domain. In contrast, ProbKT natively allows for class counts and new classes as well as other types of weak supervision.

**Neural probabilistic logical reasoning** Probabilistic logical reasoning combines logic and probability theory. Favored for its high-level reasoning abilities, it was introduced as an alternative way to deep learning in the quest for artificial intelligence [10]. Statistical artificial intelligence [32; 23] and probabilistic logic programming [11] are examples of areas relying on these premises. In a unification effort, researchers have proposed hybrid architectures, embedding both deep learning and logical reasoning components [38; 35]. Our work builds upon the recent advances in the field, where combinations of deep learning, logical, and *probabilistic* approaches were introduced [27], allowing high-level reasoning with uncertainty using differentiable neural network architectures.

| Method | Type | Annotated source dom. | Weak supervision | New classes | Implementation |
|---|---|---|---|---|---|
| Li et al. [25] | WSOD | ✗ | presence/absence | ✓ | MIL-based |
| Bilen and Vedaldi [6] | WSOD | ✗ | presence/absence | ✓ | spatial pyramid pooling layer |
| Song et al. [40] | WSOD | ✗ | presence/absence | ✓ | MIL based |
| Zhou et al. [47] | WSOD | ✗ | mix | ✓ | CAM-based |
| Bae et al. [3] | WSOD | ✗ | mix | ✓ | CAM based |
| Kundu et al. [24] | DA | ✓ | one-shot | ✓ | Class-Incremental DA |
| Saito et al. [37] | DA | ✓ | ✗ | ✗ | Strong-Weak Distribution Alignment |
| Chen et al. [8] | DA | ✓ | ✗ | ✗ | Adversarial training |
| Kim et al. [22] | DA | ✓ | ✗ | ✗ | Adversarial training and Domain Diversification |
| Zhu et al. [48] | DA | ✓ | ✗ | ✗ | selective region adaptation framework |
| Deselaers et al. [14] | WSOD w/transfer | ✓ | presence/absence | ✓ | CRF-based, iteratively |
| Zhong et al. [46] | WSOD w/transfer | ✓ | presence/absence | ✓ | MIL based, iteratively |
| Uijlings et al. [43] | WSOD w/transfer | ✓ | presence/absence | ✓ | MIL based, non iteratively |
| Inoue et al. [20] | WSOD w/transfer | ✓ | class counts | ✗ | DA + pseudolabeling, iteratively |
| ProbKT (ours) | WSOD w/transfer | ✓ | **arbitrary** | ✓ | Probabilistic logical reasoning, iteratively |

Table 1: Summary table of related works with weakly supervised object detection(WSOD), Domain Adaptation(DA) and weakly supervised knowledge transfer methods (WSOD w/ transfer).

# 3 METHODOLOGY

## 3.1 PROBLEM STATEMENT

We consider the problem of weakly supervised knowledge transfer for object detection. Using a model trained on a richly annotated source domain, we aim at improving its performance on a less richly annotated target domain.

Let $\mathcal{D}_s = \{(I_s^i, b_s^i, y_s^i) : i = 1, ..., N_s)\}$ be a dataset issued from the source domain and consisting of $N_s$ images $I_s$ along with their annotations. We write $b_s^i \in \mathbb{R}^{n_i \times 4}$ and $y_s^i \in \{1, .., K_s\}^{n_i}$ for the box coordinates and class labels of objects in image $I_s^i$. $n_i$ is the number of objects present in image $I_s^i$ and $K_s$ is the total number of object classes in the source domain. This represents the typical dataset required to train classical fully-supervised object detection architectures. The target dataset $\mathcal{D}_t = \{(I_t^i, q_t^i) : i = 1, ..., N_t)\}$ contains $N_t$ image from the target domain along with image-level annotations $q_t^i$. These annotations are logical statements about the content of the image in terms of object classes and their location. Examples include the presence of different classes in each image (*i.e.* the classical assumption in weakly supervised object detection) but also extends to the *counts* of classes or a complex combination of counts of objects attributes (*e.g.* "two red objects, and at least two bicycles"). What is more, the logical statements $q_t^i$ can include classes not already present in the source domain. This type of logical annotation is then strictly broader than the restrictive supervision usually assumed.

Based on the availability of a source dataset and a target dataset as described above, our goal is then to harness the available detailed information from the source domain to perform accurate object detection on the target domain. A graphical illustration of this process is given in Figure 1.

## 3.2 BACKGROUND

### 3.2.1 OBJECT DETECTION

Object detection aims at predicting the location and labels of objects in images. One then wishes to learn a parametric function $f_\theta : \mathcal{I} \to \{\mathcal{B} \times \mathbb{R}^K\}^{\mathbb{Z}}$ with $f_\theta(I) = \{(\hat{b}, \hat{p}_y)\}^{\hat{n}} = \{(\hat{b}_i, \hat{p}_{y,i}) : i = 1, ..., \hat{n}\}$ such that the distance between predicted and true boxes and labels, $d(\{(\hat{b}, \hat{p}_y)\}^{\hat{n}}, \{(b, y)\}^n)$, is minimum. Objects detection architecture would usually output box features proposals $\{h_i : i = 1, ..., \hat{n}\}$ conditioned on which they would predict the probability vector of class labels $\hat{p}_{y,i} = g_p(h_i)$ and the box location predictions $\hat{b}_i = g_b(h_i)$ using shared parametric functions $g_p(\cdot)$ and $g_b(\cdot)$. For an object $n$, we write the predicted probability of the object belonging to class $k$ as $\hat{p}_{y,n}^k$.

### 3.2.2 PROBABILISTIC LOGICAL REASONING

Probabilistic logical reasoning uses knowledge representation relying on probabilities that allow encoding uncertainty in knowledge. Such a knowledge is encoded in a probabilistic logical program $\mathcal{P}$ as a set of $N$ probabilistic facts $U = \{U_1, ..., U_N\}$ and $M$ logical rules $F = \{f_1, ...f_M\}$ connecting them. A simple example of probabilistic fact is "Alice and Bob will each pass their exam with probability 0.5" and an example of logical rule is "if both Alice and Bob pass their exam, they will host a party". Combining probabilistic facts and logical rules, one can then construct complex probabilistic knowledge representation, that can also be depicted as probabilistic graphical models.

Probabilistic logical programming allows to perform inference by computing the probability of a particular statement or query. For instance, one could query the probability that "Alice and Bob will host a party". This query is executed by summing over the probabilities of occurrence of the different *worlds* $w = \{u_1, ..., u_N\}$ (*i.e.* individual realization of the set of probabilistic facts) that are compatible with the query $q$. The probability of a query $q$ in a program $\mathcal{P}$ can then be inferred as $P_\mathcal{P}(q) = \sum_w P(w) \cdot \mathbb{I}[F(w) \equiv q]$, where $F(w) \equiv q$ stands for the fact that propagation of the realization $w$ across the knowledge graph, according to the logical rules $F$ leads to $q$ being true.

Remarkably, recent advances in probabilistic programming have lead to *learnable* probabilistic facts [27]. In particular, the probability of a fact can be generated by a neural network with learnable weights. Such a learnable probabilistic fact is then referred to as a neural predicate $U^\theta$, where we make the dependence on the weights $\theta$ explicit. One can then train these weights to minimize a loss that depend on the probability of a query $q$: $\hat{\theta} = \arg\min_\theta \mathcal{L}(P(q \mid \theta))$.

Our approach builds upon this ability to learn neural predicates and uses DeepProbLog [27] as the probabilistic reasoning backbone. DeepProbLog is a neural probabilistic logic programming language that allows to conveniently perform inference and differentiation with neural predicates. We refer the reader to the excellent introduction of Manhaeve et al. [28] for further details about this framework.

### 3.3 ProbKT: Weakly supervised knowledge transfer with probabilistic logical reasoning

A graphical description of our approach is presented in Figure 2. Our framework starts from a pre-trained object detection model $f_\theta$ on the source domain. The backbone of this model is extracted and inserted into a new object detection model $f_\theta^*$ with new target box position predictors and box label classifiers. This new model is then used to predict box proposals along with the corresponding box features on target domain images $I_t$. These box features are then fed to a new target box position predictor and box label classifier. The predictions of this classifier are considered neural predicates and are given to a probabilistic logical module. This module evaluates the probability of queries $q_t$, the loss, and the corresponding gradient that can be backpropagated to the classifier and the backbone. As we want to maximize the probability of the queries being true, we use the following loss function:

$$\mathcal{L}_\theta = \sum_{(I_t, q_t) \in \mathcal{D}_t} -log P_\mathcal{P}(q_t \mid f_\theta^*(I_t)) \tag{1}$$

In theory, the backbone can be trained end to end with this procedure. Our experiments showed that only updating the box features classifiers resulted in more stability as also shown in previous works [46]. We then adopt here the same iterative relabeling strategy, as described next.

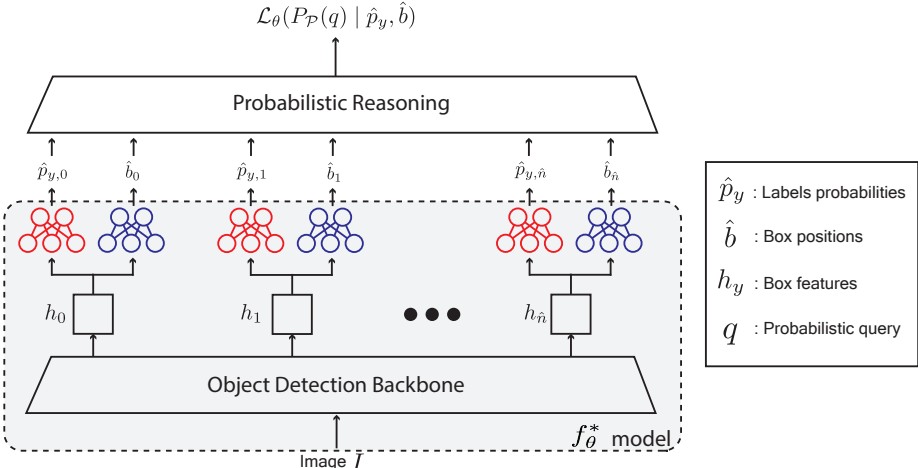

Figure 2: **ProbKT**. The pre-trained object detection backbone outputs the box features $h$ for the detected objects. Box classifiers (red) and box position predictors (blue) then predict corresponding label predictions $\hat{p}_y$ and box position predictions $\hat{b}$ that are fed to the probabilistic reasoning layer. This layer computes the probability of the query along with the gradients with respect to $\hat{p}_y$ and $\hat{b}$ that can be backpropagated through the entire network.

#### 3.3.1 Iterative relabeling

The approach described above allows to fine-tune our model $f_\theta^*$ to the target domain. To further improve the performance, we propose an iterative relabeling strategy that consists in multiple steps : fine-tuning, re-labeling and re-training. A similar has also been proposed by Zhong et al. [46].

**Fine-tuning**. This step corresponds to training ProbKT on the weakly supervised labels, by minimizing the loss of Equation 1.

**Re-labeling**. Once ProbKT has been trained, we can use its predictions to annotate images in the target domain. In practice, we only relabel images for which the model predictions comply with the available query labels in order to avoid too noisy labels.

**Re-training**. The re-labeled target domain can be used to re-train the object detection backbone of ProbKTin a fully-supervised fashion.

This procedure can be repeated multiple times to improve the quality of the relabeling and the quantity of relabelled in the target domain dataset. A graphical representation of the relabeling pipeline is presented in Figure 3.

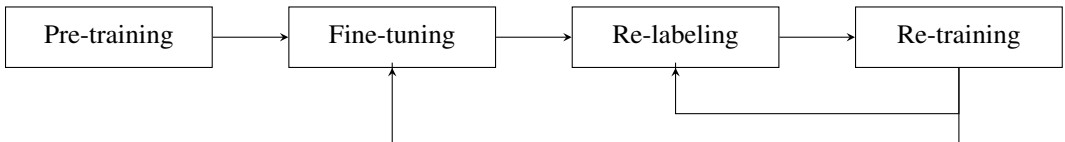

Figure 3: Iterative relabeling. A full cycle is composed of a fine-tuning, a re-labeling and a re-training step. After one cycle, the fine-tuning step and/or re-labeling step can be iteratively repeated.

### 3.3.2 COMPUTATIONAL COMPLEXITY AND APPROXIMATIONS

The computational complexity of inference in probabilistic programming depends on the specific query $q$ and several approximations have been proposed for improving the computation time [44]. We propose two approaches for reducing the computational cost adapted to object detection: (1) filtering the data samples before applying ProbKT (see Appendix Section C.1) or (2) when the supervision consists of the class labels counts, considering only the most probable world (ProbKT*) instead of all possible worlds.

### 3.3.3 PROBKT*: THE MOST PROBABLE WORLD AND CONNECTION TO HUNGARIAN MATCHING

The probabilistic inference step requires a smart aggregation of all worlds compatible with the query $q$. Yet, in certain cases, one can reduce the computational cost by only considering the most probable world. Indeed, consider the case when the query consists of the list of different class labels in the images. For a number of boxes $\hat{n}$ proposed by the objects detection model, the query can be written as the set of labels $q = \{y^i : i = 1, ..., \hat{n}\}$. If we further write $\hat{p}_{y,n}^k$ as the probability of the label of box $n$ belonging to class $k$ given by the model (as introduced in Section 3.2.1), we have:

$$P_{\mathcal{P}}(q) = \sum_{j=1}^{\hat{n}!} \hat{p}_{y,0}^{\sigma_j(0)} \cdot \hat{p}_{y,1}^{\sigma_j(1)} \cdot ... \cdot \hat{p}_{y,\hat{n}}^{\sigma_j(\hat{n})} = \sum_{j=1}^{\hat{n}!} \prod_n \hat{p}_{y,n}^{\sigma_j(n)}$$

where $\sigma_j$ corresponds to the $j^{th}$ permutation of the query vector $q$. To avoid the computation of each possible world contribution, one can only use the configuration with the largest contribution to $P_{\mathcal{P}}(q)$ and discard the other ones.

This possible world corresponds to the permutation $\sigma^*$ that satisfies:

$$\sigma^* = \arg\max_{\sigma} \log(\prod_n \hat{p}_{y,n}^{\sigma_j(n)}) = \arg\max_{\sigma} \sum_n \hat{p}_{y,n}^{\sigma_j(n)} = \arg\min_{\sigma} \sum_n (1 - \hat{p}_{y,n}^{\sigma_j(n)}).$$

Remarkably, this corresponds to the solution of the best alignment using the Hungarian matching algorithm with cost $c(n) = (1 - \hat{p}_{y,n}^{\sigma_j(n)})$, as used, among others, in DETR [7]. Thus, when the query is is the set of class labels, the most plausible world can thus be inferred with the Hungarian matching algorithm. In Appendix C.2, we also show that the gradient of ProbKT can be interpreted as a probability weighted extension of the gradient resulting from the Hungarian matching.

## 4 EXPERIMENTS

### 4.1 DATASETS

We evaluate our approach on three different datasets: (1) a CLEVR-mini dataset, (2) a Molecules dataset with images of chemical compounds, and (3) an MNIST-based object detection dataset. For each dataset, three subsets, corresponding to different domains, are used: (1) a source domain, (2) a target domain, and (3) an out-of-distribution domain (OOD). The source domain is the richly annotated domain that was used to pre-train the object detection model. The target domain is

the domain of interest but with image-level annotations only. Lastly, the OOD domain contains images from a different distribution than the source and target domains and is used to study the generalizability of the models. Source and target domains are split into 5 folds of train and validation sets and an independent test set. We focused our experiments on the small sample regime (range 1k-2k numbers of samples) both for the source as the target domain. More details on each dataset can be found in Appendix B.

## 4.2 MODELS

In the experiments, we apply our method ProbKT on two different pre-trained object detection backbone models: (1) DETR [7] and (2) FasterRCNN [34]. Both are pre-trained on the COCO dataset [26]. We also evaluate an Hungarian-algorithm approximation (ProbKT*) of our method when the weak supervision allows it. For sake of conciseness, we omit the results of ProbKT* here but they can be found in Appendix D. The details of the training procedures, as well as the hyper-parameters used for the different models and the different datasets are summarized in Table 4 in Appendix A.

### 4.2.1 BASELINE MODELS

As shown in Section 2, all available approaches for weakly supervised object detection are very restrictive in terms of the supervision signal they support. Our main comparison partner is the state of the art *WSOD-transfer* method [46].

Additionally, we compare our approach against a Resnet50 [18] backbone pre-trained on ImageNet [12]. Fine-tuning is performed by adding an extra multitask regression layer that is trained to predict the individual counts of the objects in the image as in Xue et al. [45]. This architecture naturally relies only on label counts in the target images for fine-tuning. We then predict box predictions using class activation maps as in Bae et al. [3] to compare its performance on object localization. We call this approach *Resnet50-CAM*.

When the supervision signal allows it, we also compare with a DETR model trained end-to-end jointly on target and source domains, masking the box costs in the matching cost of the Hungarian algorithm for image-level annotated samples. We call this approach *DETR-joint*.

| Model | Data Domain | CLEVR count acc. | CLEVR mAP (mAP@IoU=0.5) | Mol. count. acc | Mol. mAP (mAP@IoU=0.5) |
|---|---|---|---|---|---|
| Resnet50-CAM | target domain | $0.97 \pm 0.005$ | $0.036 \pm 0.014 \ (0.200 \pm 0.071)$ | $\mathbf{0.978 \pm 0.004}$ | $0.0 \pm 0.0 \ (0 \pm 0)$ |
| Resnet50-CAM | OOD | $0.831 \pm 0.016$ | $0.029 \pm 0.010 \ (0.153 \pm 0.044)$ | $0.0 \pm 0.0$ | n/a* |
| Resnet50-CAM | source domain | $0.993 \pm 0.003$ | $0.035 \pm 0.014 \ (0.178 \pm 0.084)$ | $0.828 \pm 0.021$ | $0.0 \pm 0.0 \ (0 \pm 0)$ |
| WSOD-transfer | target domain | $0.944 \pm 0.004$ | $0.844 \pm 0.005 \ (0.988 \pm 0.001)$ | $0.001 \pm 0.0$ | $0.018 \pm 0.004 \ (0.061 \pm 0.011)$ |
| WSOD-transfer | OOD | $0.73 \pm 0.011$ | $0.79 \pm 0.005 \ (0.969 \pm 0.001)$ | $0.003 \pm 0.002$ | n/a* |
| WSOD-transfer | source domain | $0.989 \pm 0.001$ | $0.926 \pm 0.001 \ (0.995 \pm 0.0)$ | $0.0 \pm 0.0$ | $0.021 \pm 0.003 \ (0.069 \pm 0.009)$ |
| DETR-joint | target domain | $0.159 \pm 0.133$ | $0.579 \pm 0.012 \ (0.684 \pm 0.019)$ | $0.357 \pm 0.196$ | $0.197 \pm 0.055 \ (0.481 \pm 0.071)$ |
| DETR-joint | OOD | $0.084 \pm 0.039$ | $0.534 \pm 0.012 \ (0.66 \pm 0.012)$ | $0.024 \pm 0.021$ | n/a* |
| DETR-joint | source dom. | $0.923 \pm 0.049$ | $0.908 \pm 0.017 \ (0.992 \pm 0.001)$ | $0.232 \pm 0.127$ | $0.23 \pm 0.063 \ (0.565 \pm 0.08)$ |
| RCNN (pre-trained) | target domain | $0.0 \pm 0.0$ | $0.586 \pm 0.014 \ (0.598 \pm 0.013)$ | $0.592 \pm 0.007$ | $\mathbf{0.568 \pm 0.005} \ (0.785 \pm 0.004)$ |
| RCNN (pre-trained) | OOD | $0.0 \pm 0.0$ | $0.582 \pm 0.012 \ (0.603 \pm 0.011)$ | $0.348 \pm 0.036$ | n/a* |
| RCNN (pre-trained) | source domain | $0.988 \pm 0.002$ | $\mathbf{0.984 \pm 0.01} \ (0.996 \pm 0.0)$ | $0.948 \pm 0.004$ | $\mathbf{0.737 \pm 0.005} \ \mathbf{(0.979 \pm 0.0)}$ |
| DETR (pre-trained) | target domain | $0.0 \pm 0.0$ | $0.498 \pm 0.014 \ (0.533 \pm 0.024)$ | $0.464 \pm 0.033$ | $0.314 \pm 0.006 \ (0.542 \pm 0.006)$ |
| DETR (pre-trained) | OOD | $0.0 \pm 0.0$ | $0.477 \pm 0.013 \ (0.531 \pm 0.021)$ | $0.002 \pm 0.001$ | n/a* |
| DETR (pre-trained) | source domain | $0.97 \pm 0.009$ | $0.945 \pm 0.009 \ (0.992 \pm 0.001)$ | $0.581 \pm 0.022$ | $0.409 \pm 0.005 \ (0.722 \pm 0.004)$ |
| ProbKT (DETR) | target domain | $0.946 \pm 0.014$ | $0.803 \pm 0.011 \ (0.989 \pm 0.006)$ | $0.508 \pm 0.027$ | $0.204 \pm 0.02 \ (0.507 \pm 0.014)$ |
| ProbKT (DETR) | OOD | $0.726 \pm 0.035$ | $0.715 \pm 0.006 \ (0.974 \pm 0.006)$ | $0.004 \pm 0.003$ | n/a* |
| ProbKT (DETR) | source domain | $0.987 \pm 0.003$ | $0.948 \pm 0.005 \ (0.995 \pm 0.001)$ | $0.549 \pm 0.026$ | $0.38 \pm 0.013 \ (0.713 \pm 0.006)$ |
| ProbKT (RCNN) | target domain | $\mathbf{0.975 \pm 0.003}$ | $\mathbf{0.856 \pm 0.039} \ (0.993 \pm 0.001)$ | $0.942 \pm 0.009$ | $0.289 \pm 0.041 \ \mathbf{(0.829 \pm 0.054)}$ |
| ProbKT (RCNN) | OOD | $0.89 \pm 0.022$ | $\mathbf{0.833 \pm 0.042} \ \mathbf{(0.991 \pm 0.001)}$ | $\mathbf{0.603 \pm 0.037}$ | n/a* |
| ProbKT (RCNN) | source domain | $\mathbf{0.995 \pm 0.002}$ | $0.941 \pm 0.041 \ (0.998 \pm 0.001)$ | $\mathbf{0.96 \pm 0.002}$ | $0.666 \pm 0.005 \ (0.978 \pm 0.002)$ |

Table 2: Results of the experiments for the datasets: CLEVR-mini and Molecules. Reported test accuracies over the 5 folds. Best method is in bold for each metric and data distribution. *: OOD test set of Molecules dataset has no bounding box labels.

## 4.3 EVALUATION METRICS

We evaluate the performance of the models on the different datasets based on two criteria : the count accuracy and the objects localization performance. The count accuracy measures the ratio of correct images where all individual counts of (all detected) objects are correct. To evaluate how well the

model is performing in localizing the different objects in the image we report the mean average precision (mAP) performance, a widely used metric for evaluating object detection models.

## 4.4 WEAKLY SUPERVISED KNOWLEDGE TRANSFER WITH CLASS COUNTS

We first investigate the performance of ProbKT when the weakly supervision consists of class counts only. The query $q$ for each image then consists of the number of objects from each class in the image. We evaluate the models on the CLEVR-mini and Molecules datasets. For the Molecules dataset, the query for an image containing 6 carbon atoms (C), 6 oxygen atoms (O) and 12 hydrogen atoms (H) would result in the following query: $q = ([C, O, H], [6, 6, 12])$. These weak labels in the case of the Molecules dataset are widely and easily available in the form of the chemical formula of the molecule on the image (*e.g* $C_6H_{12}O_6$). The recognition of atomic level entities on images of molecules is a challenge in the field of Optical Chemical Structure Recognition (OCSR) [9; 33; 29; 19]. For the CLEVR-mini dataset, the query for an example image containing 2 spheres, 1 cylinder and 3 cubes would be $q = ([Cube, Cylinder, Sphere], [3, 1, 2])$. Formal descriptions of the queries for each task are presented in Appendix E.

Results of the experiments are summarized in Table 2. We observe on both datasets that ProbKT is able to transfer knowledge from the source domain to the target domain and improve count accuracy on the target domain and in most cases also on the source domain. The count accuracy increases on both the target domain and on OOD, suggesting better generalization performance. This is in contrast with Resnet50-CAM which performs well on the target domain of the Molecules dataset but fails on OOD. We also note a significant improvement in object localization (mAP) for ProbKT on the CLEVR-mini dataset. However, fine-tuning seems detrimental for mAP on the Molecules dataset. This can be explained by the very small bounding boxes in the Molecules dataset. We therefore also report the mAP@IoU=0.5 where we observe some increase in performance after fine-tuning. Lastly, we observe that our approach outperforms WSOD-transfer on all metrics for both datasets. WSOD-transfer performs well on CLEVR-mini but fails for the Molecules dataset. This can be explained by the fact that this method only supports class indicators (whether a class is present in the image), which is particularly detrimental in molecules images containing a lot of objects.

## 4.5 OTHER TYPES OF WEAK SUPERVISION

### 4.5.1 CLASS RANGES

The annotation of images is a tedious task, which limits the availability of fully annotated datasets. When the number of objects on an image is large, counting the exact number of objects of a particular class becomes too time-consuming. A typical annotation in this case consists oof class *ranges* where, instead of exact class counts, an interval is given for the count. For example an image from the CLEVR-mini dataset with more than 4 cubes, exactly 4 cylinders and less than 4 spheres would result in the following query: $q = ([cube, cylinder, sphere], [[4, \infty[, [4, 5[, [0, 4[])$. We evaluate this experimental setup and report results in Table 3. We observe that ProbKT performs significantly better than WSOD-transfer on count accuracy, which still uses only presence/absence labels. We note that Resnet50-CAM is unable to use this type of supervision and is thus reported as $n/a$.

| Model | Data Domain | MNIST count acc. | MNIST sum acc. | MNIST mAP (mAP@IoU=0.5) | CLEVR* count acc. | CLEVR* mAP (mAP@IoU=0.5) |
|---|---|---|---|---|---|---|
| Resnet50-CAM | target domain | $0.044 \pm 0.041$ | $0.506 \pm 0.063$ | $0.003 \pm 0.003 (0.014 \pm 0.011)$ | n/a | n/a |
| Resnet50-CAM | OOD | $0.01 \pm 0.009$ | $0.015 \pm 0.004$ | $0.003 \pm 0.002 (0.011 \pm 0.007)$ | n/a | n/a |
| Resnet50-CAM | source domain | $0.127 \pm 0.132$ | $0.649 \pm 0.108$ | $0.005 \pm 0.004 (0.028 \pm 0.018)$ | n/a | n/a |
| WSOD-transfer | target domain | n/a | n/a | n/a | $0.944 \pm 0.004$ | $\mathbf{0.844 \pm 0.005} (0.988 \pm 0.001)$ |
| WSOD-transfer | OOD | n/a | n/a | n/a | $0.73 \pm 0.011$ | $0.79 \pm 0.005 (0.969 \pm 0.001)$ |
| WSOD-transfer | source domain | n/a | n/a | n/a | $0.989 \pm 0.001$ | $0.926 \pm 0.001 (0.995 \pm 0.0)$ |
| RCNN (pre-trained) | target domain | $0.292 \pm 0.005$ | $0.298 \pm 0.005$ | $0.632 \pm 0.014 (0.685 \pm 0.002)$ | $0.0 \pm 0.0$ | $0.586 \pm 0.014 (0.598 \pm 0.013)$ |
| RCNN (pre-trained) | OOD | $0.205 \pm 0.004$ | $0.212 \pm 0.004$ | $0.631 \pm 0.013 (0.683 \pm 0.002)$ | $0.0 \pm 0.0$ | $0.582 \pm 0.012 (0.603 \pm 0.011)$ |
| RCNN (pre-trained) | source domain | $0.961 \pm 0.008$ | $0.961 \pm 0.008$ | $\mathbf{0.917 \pm 0.021} (0.988 \pm 0.002)$ | $0.988 \pm 0.002$ | $\mathbf{0.984 \pm 0.01} (0.996 \pm 0.0)$ |
| ProbKT (RCNN) | target domain | $\mathbf{0.902 \pm 0.005}$ | $\mathbf{0.903 \pm 0.005}$ | $\mathbf{0.786 \pm 0.021} (0.974 \pm 0.001)$ | $\mathbf{0.971 \pm 0.006}$ | $0.838 \pm 0.034 (\mathbf{0.993 \pm 0.001})$ |
| ProbKT (RCNN) | OOD | $\mathbf{0.863 \pm 0.008}$ | $\mathbf{0.865 \pm 0.008}$ | $\mathbf{0.778 \pm 0.021} (0.97 \pm 0.001)$ | $\mathbf{0.884 \pm 0.01}$ | $\mathbf{0.812 \pm 0.036} (0.991 \pm 0.001)$ |
| ProbKT (RCNN) | source domain | $\mathbf{0.967 \pm 0.004}$ | $\mathbf{0.967 \pm 0.004}$ | $0.873 \pm 0.016 (0.989 \pm 0.001)$ | $\mathbf{0.994 \pm 0.001}$ | $0.922 \pm 0.035 (\mathbf{0.998 \pm 0.001})$ |

Table 3: Results of the experiments on the MNIST object detection dataset and on CLEVR* dataset (*CLEVR uses ranges of class counts as labels instead of exact class counts). Reported test accuracies over the 5 folds. Best method is in bold for each metric and data distribution.

### 4.5.2 COMPLEX QUERIES

More complex types of weak supervision than the ones considered above are also possible. To illustrate the capabilities of our approach, we build an MNIST object detection dataset where images show multiple digits as objects. Examples images are available in Appendix B. The weak supervision is here the sum of all digits in the image : $q = \texttt{SUM(digits)}$. Our ProbKT can seamlessly integrate this type of supervision as shown in Table 3. As all other baselines are unable process this type of supervision, we compare against a pre-trained RCNN and a variation of Resnet50-CAM where we add an extra neural network layer that sums the individual counts to give the resulting sum. We report count accuracy, mAP and sum accuracy. The sum accuracy measures the ratio of correct images where the predicted sum (instead of the label of the digits) is correct. Details about the results on extra experiments with DETR as backbone using complex types of weak supervision can be found in Appendix D.

### 4.6 ABLATION STUDIES

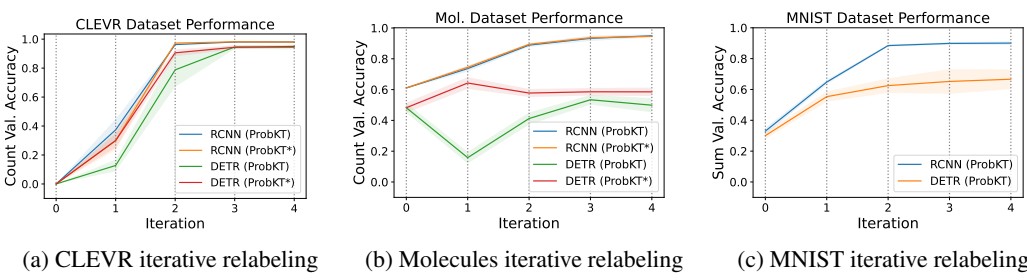

(a) CLEVR iterative relabeling     (b) Molecules iterative relabeling     (c) MNIST iterative relabeling

Figure 4: Iterative relabeling performance for the different datasets. Iteration 0: pretrained on source domain. Iteration 1: fine-tuned. Iteration 2: re-labeled and re-trained. Iteration 3: relabeled and re-trained. Iteration 4: relabeled and re-trained.

**Iterative relabeling**. In Figure 4, we plot the evolution of the performance on the test sets after multiple rounds of fine-tuning and re-labeling, as detailed in Section 3.3.1. The final performance reported in the results tables is selected based on best relabeling iteration on the validation dataset. We observe that iterative relabeling after fine-tuning can improve performance significantly. Nevertheless, the benefit of iterative relabeling is less pronounced for DETR on the Molecules dataset. We impute it to the fact that the fine-tuned DETR model is less accurate on this dataset.

**Object detection backbone**

Our method can seamlessly accommodate different object detection backbones. In Table 2, we present the results for our method with a DETR[7] and a FasterRCNN[34] backbone. We observe that FasterRCNN is typically performing better. In particular, the DETR backbone performs poorly on the Molecules dataset. This could be due to the small objects in the Molecules dataset. Indeed, Carion et al. [7] recommend to use DETR-DC5 or DETR-DC5-R101 for small objects instead.

## 5 CONCLUSIONS AND DISCUSSION

Objects detection models are a key component of machine learning deployment in the real world. However, training such models usually requires large amounts of richly annotated images that are often prohibitive for many applications. In this work, we proposed a novel approach to train object detection models by leveraging richly annotated datasets from other domains and allowing arbitrary types of weak supervision on the target domain. Our architecture relies on a probabilistic logical programming engine that efficiently blends the power of symbolic reasoning and deep learning architecture. As such, our model also inherits the current limitations from the probabilistic reasoning implementations, such as higher computational complexity. We proposed several approaches to speed-up the inference process significantly and our work will directly benefit from further advances in this field. Lastly, the versatility of probabilistic programming could help support other related tasks in the future, such as image to graph translation.

**Reproducibility Statement** Details for reproducing all experiments shown in this work are available in Appendix E. More details on the datasets used in the experiments can be found in Appendix B.

ACKNOWLEDGMENTS

AA, MO and YM are funded by (1) Research Council KU Leuven: Symbiosis 4 (C14/22/125), Symbiosis3 (C14/18/092); (2) Federated cloud-based Artificial Intelligence-driven platform for liquid biopsy analyses (C3/20/100); (3) CELSA - Active Learning (CELSA/21/019); (4) European Union's Horizon 2020 research and innovation programme under the Marie Skłodowska-Curie grant agreement No. 956832; (5) Flemish Government (FWO: SBO (S003422N), Elixir Belgium (I002819N), SB and Postdoctoral grants: S003422N, 1SB2721N, 1S98819N, 12Y5623N) and (6) VLAIO PM: Augmenting Therapeutic Effectiveness through Novel Analytics (HBC.2019.2528); (7) YM, AA, EDB, and MO are affiliated to Leuven.AI and received funding from the Flemish Government (AI Research Program). EDB is funded by a FWO-SB grant (S98819N). Computational resources and services used in this work were partly provided by the VSC (Flemish Supercomputer Center), funded by the Research Foundation - Flanders (FWO) and the Flemish Government – department EWI.

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

# A    TRAINING DETAILS

For the hyper-parameters the idea was to stay as close as possible to the defaults of the pre-trained standard models although some lightweight tuning was done. In Table 4 a summary is given for the hyper-parameters used for the different models.

| Model | dataset | epochs | lr | lr_step_size | lr-gamma | momentum | batch size | weight decay | optimizer |
|---|---|---|---|---|---|---|---|---|---|
| DETR pre-train (retrain) | CLEVR | max 100 | 0.0001 | 7 (7-8) | 0.1 | | 8 | 0.0001 | AdamW |
| DETR pre-train (retrain) | Mols. | max 100 | 0.0001 | 20 (20) | 0.1 | | 8 | 0.0001 | AdamW |
| DETR pre-train (retrain) | MNIST | max 100 | 0.0001 | 15-20 (20) | 0.1 | | 8 | 0.0001 | AdamW |
| RCNN pre-train (retrain) | all datasets | max 30 | 0.005 | 5 (5) | 0.1 | 0.9 | 1 | 0.0005 | SGD |
| RCNN Finetune | all datasets | max 20 | 0.001 | | | | 16 | | Adam |
| DETR Finetune | CLEVR/Mols | max 20 | 0.001 | | | | 16 | | Adam |
| DETR Finetune | MNIST | max 20 | 0.01 | | | | 16 | | Adam |
| DETR Finetune* | CLEVR/Mols | max 100 | 0.002 | 20 | 0.1 | | 8 | 0.0001 | AdamW |
| RCNN Finetune* | CLEVR | max 20 | 0.001 | | | | 15 | | Adam |
| RCNN Finetune* | Mols | max 20 | 0.00001 | | | | 15 | | Adam |
| DETR masked box loss | CLEVR/Mols | max 100 | 0.0001 | 7 | 0.1 | | 8 | 0.0001 | AdamW |
| Resnet50-CAM  models | all datasets | max 500 | 0.001 | | | | 32 | | Adam |

Table 4: Overview of hyperparameters for the different models, most hyperparamaters are left default from standard models. Tuning was mostly done on learning rate and learning rate scheduling. For every fold/dataset the best epoch/lr/lr_step_size model is used based on validation data.

# B    DATASETS

We evaluate our approach on three different datasets: (1) a CLEVR-mini dataset, (2) a Molecules data set with images of chemical compounds, and (3) an MNIST-based object detection dataset. For each dataset, three subsets, corresponding to different domains, are used: (1) a source domain, (2) a target domain, and (3) an out-of-distribution domain (OOD). Source and target domains are split into 5 folds of train and validation sets and an independent test set. Sizes of the different splits per dataset are summarized in Table 5.

| Dataset | Type | Split | Size (number of samples) |
|---|---|---|---|
| MNIST object detection | Source | train | 700 |
| MNIST object detection | Source | validation | 300 |
| MNIST object detection | Source | test | 1000 |
| MNIST object detection | Target | train | 700 |
| MNIST object detection | Target | validation | 300 |
| MNIST object detection | Target | test | 1000 |
| MNIST object detection | OOD | test | 1000 |
| Molecules | Source | train | 1400 |
| Molecules | Source | validation | 600 |
| Molecules | Source | test | 1000 |
| Molecules | Target | train | 1400 |
| Molecules | Target | validation | 600 |
| Molecules | Target | test | 1000 |
| Molecules | OOD | test | 1000 |

Table 5: Dataset sizes for the different splits. For train and validations splits 5 folds are used.

### B.0.1   CLEVR-MINI DATASET

The CLEVR-mini dataset for our experiments is a selection of samples from the CLEVR dataset [21]. The different types available in the CLEVR dataset are combinations of shapes (cube, sphere, and cylinder), materials (metal and rubber), and sizes (large and small). Colors are ignored as the images are first converted to grayscale before feeding them to the models. For the richly annotated source domain, we randomly select images with only sphere or cylinder-shaped objects (no cubes) and with a maximum of four objects per image and a minimum of three objects. For the weakly annotated target domain we experiment with two type of annotations. Firstly we experiment when we have the class counts of objects in the image available. Secondly, instead of the exact counts of classes in the image the annotations only specify if there is exactly one object class in the image or multiple. The advantage of this kind of labeling is that the annotator does not need to count the objects and instead just make a distinction of only one object class in image or multiple. The images in the target domain can contain all combinations of object types (including cube-shaped objects) and allow a minimum of five objects per image and a maximum of six objects per image. For the OOD dataset we also select images with all possible combinations of object types, always with 10 objects per image. Some example images from the CLEVR-mini dataset can be found in Figure 1.

### B.0.2   MOLECULES DATASET

The Molecules dataset contains images depicting chemical compounds. For the richly annotated source domain, a procedure similar as described in Oldenhof et al. [29, 30] was executed using an RDKit [2] fork for generating the bounding box labels for the individual atoms present in the images. In the source domain, we allow the following atom types: carbon (C), hydrogen (H), oxygen (O), and nitrogen (N). In the weakly annotated target domain, we only have the counts of the atoms present which translates to the chemical formula of the molecule in the image (e.g $C_6H_{12}O_6$). The same classes from the source domain (C, H, O, and N) are also present in the target domain as well as an extra atom type: sulfur (S). The OOD test dataset consists of 1000 images from the external UoB dataset [36] where chemical compounds containing only the atom types present in the target domain (C, H, O, N, and S). Some example images from the Molecules dataset are visualized in Figure 5.

### B.0.3   MNIST OBJECT DETECTION DATASET

The MNIST object detection dataset is generated [1] using the original MNIST dataset [13]. Each image consists of three MNIST digits randomly positioned in the image. The MNIST object detection dataset allows experimenting with a more arbitrary type of weak supervision. Each object in this dataset represents a digit that can be aggregated. This allows to label an image with only the *sum* of all digits in the image instead of the class counts of the objects. For the richly annotated source domain digits 7, 8, and 9 are left out. The weakly annotated target domain has all possible digit classes (0-9). The labels of the target domain only contain the sum of all digits. For the OOD test dataset, images are used that contain maximum of four MNIST digits, instead of three digits as in the other domains. Some example images from the MNIST object detection dataset are visualized in Figure 6.

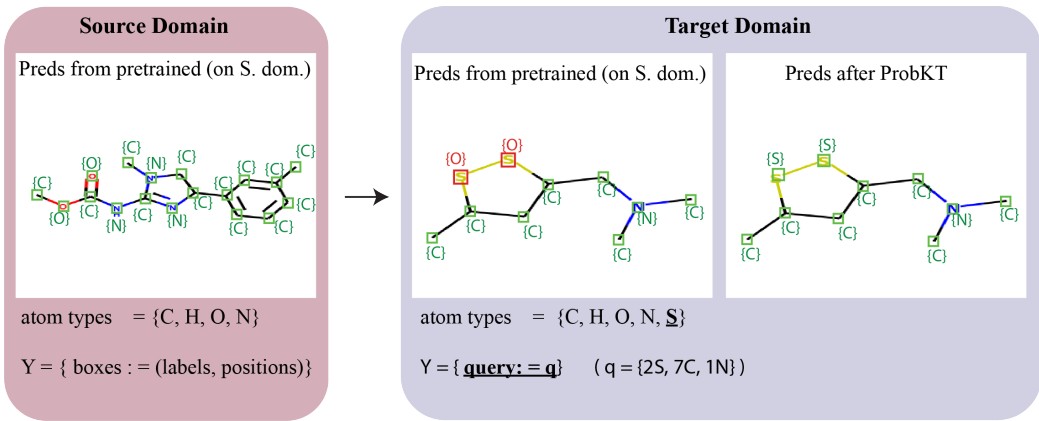

Figure 5: Weakly supervised knowledge transfer with probabilistic logical reasoning (ProbKT). On the left we have source domain where a model can be trained using bounding box information(labels,positions) but only on a limited set of atom types (C,H, O, N). In the middle we can see that the pre-trained model is not able to recognize the sulfur (S) from target domain correctly. On the right we see that the model is able to adapt to target domain after probabilistic reasoning using weak labels (e.g. counts of objects on image) and is able to recognize the sulfur (S).

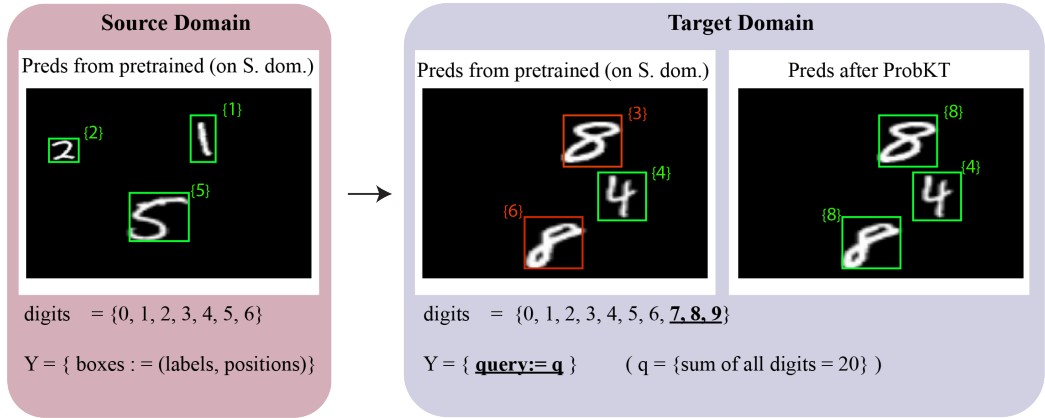

Figure 6: Weakly supervised knowledge transfer with probabilistic logical reasoning (ProbKT). On the left we have source domain where a model can be trained using bounding box information(labels,positions) but only on a limited set of digits (0, 1, 2, 3, 4, 5, 6). In the middle we can see that the pre-trained model is not able to recognize the digit eight (8) from target domain correctly. On the right we see that the model is able to adapt to target domain after probabilistic reasoning using weak labels (e.g. sum of digits on image) and is able to recognize the digit eight (8).

## C  PROBKT AND PROBKT* SUPPLEMENTARY DETAILS

### C.1  FILTERING SAMPLES

The computation complexity of inference in the probabilistic programming module grows with the number of possible worlds. In turn, the number of possible worlds grows with the number of probabilistic facts $\hat{n}$.

One avenue to reduce the computational cost of the inference step is then to artificially reduce the number of probabilistic facts in each image. Let $\{\hat{p}_{y,n} : n = 1, ..., \hat{n}\}$ and $q$ the corresponding inference query. We compute the filtered set of probabilistic facts $\bar{p}_{y,n}$ by setting

$$\bar{p}_{y,n}^k = \begin{cases} 1 & \text{if } \hat{p}_{y,n}^k \geq \delta \\ 0 & \text{if } \exists k' \text{ s.t. } \hat{p}_{y,n}^{k'} \geq \delta \quad \text{and} \quad \hat{p}_{y,n}^k < \delta \\ \hat{p}_{y,n}^k & \text{otherwise.} \end{cases} \tag{2}$$

The parameter $\delta \in [0,1]$ is a threshold at which we consider the probabilistic fact as certain. A probability of 1 or 0 effectively discards the probabilistic fact $\bar{p}_{y,n}$ from the inference procedure. However, we also have to update the inference query $q$ to reflect this filtration. We write $\bar{q}$ the filtered query $q$.

**Example** To illustrate this filtration strategy let's consider an MNIST image with 3 digits in the image : $\{3, 4, 7\}$. The query $q$ corresponds to the class labels in the images. That is $q = \{3, 4, 7\}$. The object detection backbones outputs 3 box features with corresponding probabilities $\{\hat{p}_{y,0}, \hat{p}_{y,1}, \hat{p}_{y,2}, \}$. Now let *e.g.* $\hat{p}_{y,1}^3 = 0.99$. We can filter out $\hat{p}_{y,1}$ (*i.e* the prediction for a digit 3 is certain), and compute the filtered query $\bar{q} = \{4, 7\}$.

**Remark** Equation 2 suggests a filtering based on the output probabilities only. However, one can also use information about the query for the filtration. For instance, one would only filter out a probabilistic fact if it is consistent with the query $q$. In the example above, it would be wiser not to filter out *e.g.* $\hat{p}_{y,1}^9 = 0.99$ as no images are supposedly present in the image. One should then ideally propagate this probabilistic fact to the inference module such as to update the weights of the backbone and learn from this error.

## C.2 GRADIENT OF THE LIKELIHOOD

The ProbKT likelihood has the following form:

$$P_{\mathcal{P}}(q) = \sum_{\alpha \in E_q} \prod_i \prod_j \hat{p}_{ij}^{\alpha_{ij}},$$

where $\alpha$ is a "possible world" matrix of indicator variables:

$$\alpha_{ij} = \begin{cases} 1 & \text{object i is of class j} \\ 0 & \text{otherwise,} \end{cases}$$

and $E_q$ is the set of all possible $\alpha$ worlds compatible with the logical annotation $q$.

**Lemma 1.** *The gradient of the likelihood has the following form:*

$$\frac{\partial P_{\mathcal{P}}(q)}{\partial \theta} = \sum_i \sum_j \frac{\partial p_{ij}}{\partial \theta} C_{ij},$$

*where the weight has the form:*

$$C_{ij} = P(E|O_i = j) = \sum_{\alpha \in E|_{O_i = j}} \prod_{i'} \prod_{j'} I_{(i \neq i' \vee j \neq j')} p_{ij}^{\alpha_{ij}}$$

In case of the Hungarian matching the most probable possible word is selected, which corresponds to setting the conditional probability $P(E|O_i = j)$ to 1 if object $i$ is paired with label $j$ and 0 otherwise. The ProbKT gradient can be interpreted as a probability weighted extension of the gradient resulting from the Hungarian matching.

## D FULL RESULTS

In Table 6, we present the full results for the MNIST experiment. We report the count accuracy (*i.e.*. correct identification of the digits in the image), sum accuracy (*i.e.* correct estimation of the sum of

| Model | Type | mnist count acc. | mnist sum acc. | mnist mAP (mAP@IoU=0.5) |
|---|---|---|---|---|
| Resnet50-CAM (baseline) | In-distribution | $0.044 \pm 0.041$ | $0.506 \pm 0.063$ | $0.003 \pm 0.003 (0.014 \pm 0.011)$ |
| Resnet50-CAM (baseline) | OOD | $0.01 \pm 0.009$ | $0.015 \pm 0.004$ | $0.003 \pm 0.002 (0.011 \pm 0.007)$ |
| Resnet50-CAM (baseline) | Source Domain | $0.127 \pm 0.132$ | $0.649 \pm 0.108$ | $0.005 \pm 0.004 (0.028 \pm 0.018)$ |
| DETR (Pre-trained) | In-distribution | $0.26 \pm 0.012$ | $0.262 \pm 0.01$ | $0.518 \pm 0.014 (0.637 \pm 0.017)$ |
| DETR (Pre-trained) | OOD | $0.173 \pm 0.01$ | $0.177 \pm 0.009$ | $0.51 \pm 0.012 (0.632 \pm 0.015)$ |
| DETR (Pre-trained) | Source Domain | $0.859 \pm 0.031$ | $0.86 \pm 0.031$ | $0.781 \pm 0.009 (0.957 \pm 0.008)$ |
| DETR (ProbKT) | In-distribution | $0.662 \pm 0.064$ | $0.664 \pm 0.065$ | $0.615 \pm 0.025 (0.856 \pm 0.037)$ |
| DETR (ProbKT) | OOD | $0.532 \pm 0.083$ | $0.533 \pm 0.082$ | $0.591 \pm 0.03 (0.845 \pm 0.038)$ |
| DETR (ProbKT) | source domain | $0.878 \pm 0.023$ | $0.879 \pm 0.023$ | $0.737 \pm 0.014 (0.952 \pm 0.009)$ |
| RCNN (Pre-trained) | In-distribution | $0.292 \pm 0.005$ | $0.298 \pm 0.005$ | $0.632 \pm 0.014 ( 0.685 \pm 0.002 )$ |
| RCNN (Pre-trained) | OOD | $0.205 \pm 0.004$ | $0.212 \pm 0.004$ | $0.631 \pm 0.013 ( 0.683 \pm 0.002)$ |
| RCNN (Pre-trained) | source domain | $0.961 \pm 0.008$ | $0.961 \pm 0.008$ | $0.917 \pm 0.021 (0.988 \pm 0.002)$ |
| RCNN (ProbKT) | In-distribution | $0.902 \pm 0.005$ | $0.903 \pm 0.005$ | $0.786 \pm 0.021 (0.974 \pm 0.001)$ |
| RCNN (ProbKT) | OOD | $0.863 \pm 0.008$ | $0.865 \pm 0.008$ | $0.778 \pm 0.021 (0.97 \pm 0.001)$ |
| RCNN (ProbKT) | source domain | $0.967 \pm 0.004$ | $0.967 \pm 0.004$ | $0.873 \pm 0.016 (0.989 \pm 0.001)$ |

Table 6: Results of the SUM experiments on the MNIST object detection dataset. Reported test accuracies over the 5 folds.

the digits in the image) and the mean average precision (mAP) (*i.e.* a common object detection metric that reflects the ability to predict the positions and labels of the objects). We observe that the Resnet baseline performs poorly, lacking the necessary logic to process this dataset. We used both DETR and RCNN as object detection backbones in our experiments, showing high test accuracies when fine-tuned with our approach. As the results suggest, RCNN backbones lead to better performance than the DETR backbone.

| Model | Data Domain | CLEVR count acc. | CLEVR mAP (mAP@IoU=0.5) | Mol. count. acc | Mol. mAP (mAP@IoU=0.5) |
|---|---|---|---|---|---|
| Resnet50-CAM | target domain | $0.97 \pm 0.005$ | $0.036 \pm 0.014 (0.200 \pm 0.071)$ | $\mathbf{0.978 \pm 0.004}$ | $0.0 \pm 0.0 (0 \pm 0)$ |
| Resnet50-CAM | OOD | $0.831 \pm 0.016$ | $0.029 \pm 0.010 (0.153 \pm 0.044)$ | $0.0 \pm 0.0$ | n/a[1] |
| Resnet50-CAM | source domain | $0.993 \pm 0.003$ | $0.035 \pm 0.019 (0.178 \pm 0.084)$ | $0.828 \pm 0.021$ | $0.0 \pm 0.0 (0 \pm 0)$ |
| WSOD-transfer | target domain | $0.944 \pm 0.004$ | $0.844 \pm 0.005 (0.988 \pm 0.001)$ | $0.001 \pm 0.0$ | $0.018 \pm 0.004 (0.061 \pm 0.011)$ |
| WSOD-transfer | OOD | $0.73 \pm 0.011$ | $0.79 \pm 0.005 (0.969 \pm 0.001)$ | $0.003 \pm 0.002$ | n/a[1] |
| WSOD-transfer | source domain | $0.989 \pm 0.001$ | $0.926 \pm 0.001 (0.995 \pm 0.0)$ | $0.0 \pm 0.0$ | $0.021 \pm 0.003 (0.069 \pm 0.009)$ |
| DETR-joint | target domain | $0.159 \pm 0.133$ | $0.579 \pm 0.012 (0.684 \pm 0.019)$ | $0.357 \pm 0.196$ | $0.197 \pm 0.055 (0.481 \pm 0.071)$ |
| DETR-joint | OOD | $0.084 \pm 0.039$ | $0.534 \pm 0.012 (0.66 \pm 0.012)$ | $0.024 \pm 0.021$ | n/a[1] |
| DETR-joint | source dom. | $0.923 \pm 0.049$ | $0.908 \pm 0.017 (0.992 \pm 0.001)$ | $0.232 \pm 0.127$ | $0.23 \pm 0.063 (0.565 \pm 0.08)$ |
| DETR (Pre-trained) | target domain | $0.0 \pm 0.0$ | $0.498 \pm 0.019 (0.533 \pm 0.024)$ | $0.464 \pm 0.033$ | $0.314 \pm 0.006 (0.542 \pm 0.006)$ |
| DETR (Pre-trained) | OOD | $0.0 \pm 0.0$ | $0.477 \pm 0.013 (0.531 \pm 0.021 )$ | $0.002 \pm 0.001$ | n/a[1] |
| DETR (Pre-trained) | source domain | $0.97 \pm 0.009$ | $0.945 \pm 0.009 (0.992 \pm 0.001)$ | $0.581 \pm 0.022$ | $0.409 \pm 0.005 (0.722 \pm 0.004)$ |
| ProbKT*(DETR) | target domain | $0.949 \pm 0.005$ | $0.728 \pm 0.014 (0.99 \pm 0.003)$ | $0.589 \pm 0.042$ | $0.373 \pm 0.02 (0.669 \pm 0.045)$ |
| ProbKT*(DETR) | OOD | $0.741 \pm 0.038$ | $0.606 \pm 0.017 (0.977 \pm 0.004)$ | $0.008 \pm 0.008$ | n/a[1] |
| ProbKT*(DETR) | source domain | $0.985 \pm 0.004$ | $0.937 \pm 0.006 (0.995 \pm 0.001)$ | $0.275 \pm 0.066$ | $0.371 \pm 0.021 (0.649 \pm 0.041)$ |
| ProbKT(DETR) | target domain | $0.946 \pm 0.014$ | $0.803 \pm 0.011 (0.989 \pm 0.006)$ | $0.508 \pm 0.027$ | $0.204 \pm 0.02 (0.507 \pm 0.014)$ |
| ProbKT(DETR) | OOD | $0.726 \pm 0.035$ | $0.715 \pm 0.006 (0.974 \pm 0.006)$ | $0.004 \pm 0.003$ | n/a[1] |
| ProbKT(DETR) | source domain | $0.987 \pm 0.003$ | $0.948 \pm 0.005 (0.995 \pm 0.001)$ | $0.549 \pm 0.026$ | $0.38 \pm 0.013 (0.713 \pm 0.006)$ |
| RCNN (pre-trained) | target domain | $0.0 \pm 0.0$ | $0.586 \pm 0.014 (0.598 \pm 0.013)$ | $0.592 \pm 0.007$ | $\mathbf{0.568 \pm 0.005} (0.785 \pm 0.004)$ |
| RCNN (pre-trained) | OOD | $0.0 \pm 0.0$ | $0.582 \pm 0.012 (0.603 \pm 0.011)$ | $0.348 \pm 0.036$ | n/a[1] |
| RCNN (pre-trained) | source domain | $0.988 \pm 0.002$ | $\mathbf{0.984 \pm 0.01} (0.996 \pm 0.0)$ | $0.948 \pm 0.004$ | $\mathbf{0.737 \pm 0.005} (\mathbf{0.979 \pm 0.0})$ |
| ProbKT*(RCNN) | target domain | $0.974 \pm 0.004$ | $0.855 \pm 0.025 (\mathbf{0.994 \pm 0.001})$ | $0.945 \pm 0.006$ | $0.24 \pm 0.042 (0.788 \pm 0.073)$ |
| ProbKT*(RCNN) | OOD | $\mathbf{0.901 \pm 0.017}$ | $0.827 \pm 0.022 (\mathbf{0.991 \pm 0.001})$ | $0.592 \pm 0.032$ | n/a[1] |
| ProbKT*(RCNN) | source domain | $0.993 \pm 0.002$ | $0.95 \pm 0.021 (\mathbf{0.998 \pm 0.0})$ | $0.96 \pm 0.003$ | $0.655 \pm 0.01 (0.974 \pm 0.004)$ |
| ProbKT(RCNN) | target domain | $\mathbf{0.975 \pm 0.003}$ | $\mathbf{0.856 \pm 0.039} (0.993 \pm 0.001)$ | $0.942 \pm 0.009$ | $0.289 \pm 0.041 (\mathbf{0.829 \pm 0.054})$ |
| ProbKT(RCNN) | OOD | $0.89 \pm 0.022$ | $\mathbf{0.833 \pm 0.042} (\mathbf{0.991 \pm 0.001})$ | $\mathbf{0.603 \pm 0.037}$ | n/a[1] |
| ProbKT(RCNN) | source domain | $\mathbf{0.995 \pm 0.002}$ | $0.941 \pm 0.041 (0.998 \pm 0.001)$ | $\mathbf{0.96 \pm 0.002}$ | $0.666 \pm 0.005 (0.978 \pm 0.002)$ |

Table 7: Results of the experiments for the datasets: CLEVR-mini and Molecules. Reported test accuracies over the 5 folds. Best method is in bold for each metric and data distribution.

---

[1] OOD test set of Molecules dataset has no bounding box labels.

## E   Source code and datasets

The source code and basic instructions are available on `https://github.com/molden/ProbKT`. The source code integrates features from the Weights & Biases (WandB) platform [5]. Basic features are supported without the need for an account on WandB but to make full use of all features we recommend to create an account.

Datasets can be downloaded here:

- CLEVR-mini dataset `https://figshare.com/s/db012765e5a38e14ef9c`
- Molecules dataset `https://figshare.com/s/3dc3508d39bf4cff8c7f`
- MNIST object detection dataset `https://figshare.com/s/c760de026f000524db5a`

**ProbLog script used in the ProbKT Probabilistic logical reasoning framework for counting of objects on an image (as on CLEVR-mini dataset):**

```
:- use_module(library(lists)).
nn(mnist_net,[X],Y,[0,1,2,3,4,5,6,7,8,9,10,11]) :: digit(X,Y).

count([],X,0).
count([X|T],X,Y):- count(T,X,Z), Y is 1+Z.
count([X1|T],X,Z):- X1\=X,count(T,X,Z).

countall(List,X,C) :-
    sort(List,List1),
    member(X,List1),
    count(List,X,C).

roll([],L,L).
roll([H|T],A,L):- roll(T,[Y|A],L), digit(H,Y).

countpart(List,[],[]).
countpart(List,[H|T],[F|L]):- countall(List,H,F), countpart(List,T,L).

count_objects(X,L,C) :- roll(X,[],Result), countpart(Result,L,C).
```

The query $q$ in the case of class counts would be `count_objects(X,L,C)`. For example an image $X$ with 1 small metal cube and 3 large rubber cylinders would result in the following query: `count_objects(X,[small_metal_cube,large_rubber_cylinder],[1,3])`.

**ProbLog script used in the ProbKT Probabilistic logical reasoning framework for aggregating the digits on an image:**

```
:- use_module(library(lists)).
nn(mnist_net,[X],Y,[0,1,2,3,4,5,6,7,8,9]) :: digit(X,Y).

sum([],0).
sum([X|T],Y):- sum(T,Z), Y is X+Z.

roll([],L,L).
roll([H|T],A,L):- roll(T,[Y|A],L), digit(H,Y).

sum_digits(X,Y) :- roll(X,[],Result), sum(Result,Y).
```

The query $q$ in the case of sum of digits would be `sum_digits(X,Y)`. For example an image $X$ with as sum of digits 12 would result in the following query: `sum_digits(X,12)`.

**ProbLog script used in the ProbKT Probabilistic logical reasoning framework for taking into account non-exact counts on images**

```
:- use_module(library(lists)).
nn(mnist_net,[X],Y,[0,1,2,3,4,5,6,7,8,9,10,11]) :: digit(X,Y).

count([],X,0).
count([X|T],X,Y):- count(T,X,Z), Y is 1+Z.
count([X1|T],X,Z):- X1\=X,count(T,X,Z).

countall(List,X,C,A) :-
    A=0,
    sort(List,List1),
    member(X,List1),
    count(List,X,C).
countall(List,X,C,A) :-
    A=1,
    sort(List,List1),
    member(X,List1),
    count(List,X,R),
    R>C.
countall(List,X,C,A) :-
    A=-1,
    sort(List,List1),
    member(X,List1),
    count(List,X,R),
    R<C.

roll([],L,L).
roll([H|T],A,L):- roll(T,[Y|A],L), digit(H,Y).

countpart(List,[],[],[]).
countpart(List,[H|T],[F|L],[A|B]):- countall(List,H,F,A),
countpart(List,T,L,B).

range_count_objects(X,L,C,S) :- roll(X,[],Result),
countpart(Result,L,C,S).
```

The query $q$ in the case of non exact counts of objects would be `range_count_objects(X,L,C,S)`. For example an image $X$ with exactly one metal small cube and multiple rubber large spheres would result in the following query: `range_count_objects(X,[s_metal_cube,l_rubber_sphere],[1,1],[0,1])`.

### E.1 INFERENCE EXAMPLE FOR MNIST DATASET

To illustrate the inference process let us follow the evaluation of the clause `sum([x1, x2],8)`, what can result from query `sum_digits(X,8)` in case of two visible digit in the image X.

This clause is true if and only if $X_1 + X_2 = 8$.

In case of MNIST digits ($\{0,1,\ldots,9\}$) enumerating the possible worlds would give the following set:

$$\{(0,8),(1,7),(2,6),\ldots,(8,0)\} \tag{3}$$

After summing the probability of all possible worlds we get:

$$p_1(0)p_2(8) + p_1(1)p_2(7) + \cdots + p_1(0)p_2(8), \tag{4}$$

where $p_1$ and $p_2$ are the distribution of random variable $X_1$ and $X_2$ respectively.

Or in a general form:

$$p_Y(Y) = \sum_{X_1} p_1(X_1)p_2(Y - X_1). \tag{5}$$

As expected the distribution of the sum is the convolution of the distributions of the two terms. This observation trivially generalizes to more than two terms. The cost function corresponding to the maximum likelihood estimation is the negative log-likelihood $-\log(p_Y(Y))$.

