# OpenReview forum: "Weakly Supervised Knowledge Transfer with Probabilistic Logical Reasoning for Object Detection"
_ICLR.cc/2023/Conference — ICLR 2023 poster_

### Official Review · Reviewer_tNrs · 2022-10-22

**Confidence:** 3
**Correctness:** 3
**Technical Novelty And Significance:** 3
**Empirical Novelty And Significance:** 3
**Recommendation:** 8

**Clarity, Quality, Novelty And Reproducibility:**

**Novelty:**

The paper seems novel and relevant.

**Clarity:**

The paper could improve in its writing.
Some things could be:
    1. Introduce better DeepProbLog and clarify how the model relies on it
    2. Move the related work, right now it is in the middle of the paper

**Reproducibility:**

The authors have made the code available.

**Future work suggestion:**

A dataset for action detection in autonomous driving has just been released with a broad set of propositional logic constraints in [1]. Maybe in the future this framework could be applied in the autonomous driving field.

**Minor suggestion:**
- specify the query $q$ in the image

[1]  Eleonora Giunchiglia, Mihaela Catalina Stoian, Salman Khan, Fabio Cuzzolin, Thomas Lukasiewicz. ROAD-R: The Autonomous Driving Dataset with Logical Requirements. Machine Learning, 2022.

**Strength And Weaknesses:**

**Strengths:**

1) The paper addresses the important problem of training object detection models with weak supervisions
2) It is the first paper to allow for more complex weak supervisions
3) The experimental analysis is comprehensive and well documented

**Weaknesses:**

1) The authors seem to be sweeping under the rug their reliance on DeepProbLog. Indeed, if I understood correctly, the reasoning and backprogation is done through the DeepProbLog framework. If this indeed the case then it needs to be made clearer and a short introduction on the framework should be given.
2) It is not clear the exact expressivity admitted by the weak supervision, is it only limited to rules? Is it full first order logic?



**Summary Of The Paper:**

The paper presents a new way of doing object detection with weak supervisions. The model proposed is called ProbKT and it allows any object detection model to be trained on a general and rich source domain, and then transfer the acquired knowledge on a weakly annotated target domain. The novelty of this work lies in the fact that it allows to train object detection models with arbitrary types of weak supervisions.

**Summary Of The Review:**

**TL;DR:** the paper seems novel and relevant. however, its relationship with deepproblog needs to be clarified

---

> ### Author Response · Authors · 2022-11-15
> **Answers to reviewer's comments**
>
> We thank reviewer tNrs for their insightful feedback that helped us improve our paper. Please find our answers below.
>
> ### About the reliance on DeepProbLog
> `The authors seem to be sweeping under the rug their reliance on DeepProbLog. Indeed, if I understood correctly, the reasoning and backpropagation is done through the DeepProbLog framework. If this indeed the case then it needs to be made clearer and a short introduction on the framework should be given.`
>
> We thank the reviewer for this suggestion and we want to make sure we acknowledge properly the reliance of our framework on DeepProbLog. We clarified this in Section 3.2.2 of the paper by adding a paragraph discussing DeepProbLog. For convenience, we copied the excerpt below. Nevertheless, we want to emphasize that our architecture is not bound to a specific probabilistic reasoning engine (in the same way that one can train a deep-learning architecture with different auto-differentiation packages). In particular, ProbKT*, the variant of our main model, uses a relaxation of the probabilistic inference which relies on the Hungarian algorithm and therefore does not require DeepProbLog.
>
> Excerpt of the updated version of the paper where we discuss our reliance on DeepProbLog:
> “Our approach builds upon this ability to learn neural predicates and uses DeepProbLog [1] as the probabilistic reasoning backbone. DeepProbLog is a neural probabilistic logic programming language that allows to conveniently perform inference and differentiation with neural predicates. We refer the reader to the excellent introduction of Manhaeve et al. [2] for further details about this framework.”
>
> References :
>
> [1] Robin Manhaeve, Sebastijan Dumancic, Angelika Kimmig, Thomas Demeester, and Luc De Raedt. Deepproblog: Neural probabilistic logic programming. Advances in Neural Information Processing Systems, 31, 2018.
>
> [2] Robin Manhaeve, Sebastijan Dumancic, Angelika Kimmig, Thomas Demeester, and Luc De Raedt. Neural probabilistic logic programming in deepproblog. Artificial Intelligence, 298: 103504, 2021
>
> ### About the expressivity of the method
> `It is not clear the exact expressivity admitted by the weak supervision, is it only limited to rules? Is it full first order logic?`
>
> ProbKT inherits the expressiveness of Prolog[1] which is based on a subset of first-order predicate logic, Horn clauses and which is Turing-complete. We clarified this in the related work section (Section 2) of the paper.
> [1] Leon Sterling and Ehud Y Shapiro. The art of Prolog: advanced programming techniques. MIT press, 1994.
>
> ### About the related work section
> `Move the related work, right now it is in the middle of the paper`
>
> We have now moved the related work section after the introduction to help the reader situate our contribution in the existing literature.
>
> ### About suggestions for another dataset
>
> `A dataset for action detection in autonomous driving has just been released with a broad set of propositional logic constraints. Maybe in the future this framework could be applied in the autonomous driving field.`
>
> We thank the reviewer for their recommendation for future work. This is indeed a very interesting application that also underlines the pertinence of our approach. For sake of completeness, we have added the citation of the dataset in the introduction to support our argumentation regarding real-world applications.
>
> ### About having the queries q in the images.
> `specify the query q in the image`
>
> This is a very nice suggestion and we have now updated our figure with the specific queries.

---

> > ### Comment · Reviewer_tNrs · 2022-11-30
> > **Thanks for the answers**
> >
> > First of all I would like to thank the authors for their clarifications, and I can also see that the paper writing has improved a lot.
> > I have updated my score accordingly.

---

### Official Review · Reviewer_q7Gr · 2022-10-24

**Confidence:** 3
**Correctness:** 4
**Technical Novelty And Significance:** 3
**Empirical Novelty And Significance:** Not applicable
**Recommendation:** 8

**Clarity, Quality, Novelty And Reproducibility:**

Clarity: 8/10, paper is well written.

Quality: 7/10.

Novelty: 7/10.

Reproducibility: 9/10, code is released.


**Strength And Weaknesses:**

# Strengths
+ By formulating the weakly supervised object detection as the probabilistic logical reasoning, a unified detection method can be trained on dataset with different supervision signals, which is a novel approach for weakly supervised knowledge transfer.
+ The paper presentation is clear, with well-defined terms and concepts.
+ Ablation studies are thorough. The effects of proposed components are validated on several datasets.
# Weaknesses
- It would be great if the method can be evaluated on datasets with real world objects like COCO or LVIS.


**Summary Of The Paper:**

This paper introduces a new method for weakly supervised object detection with knowledge transfer, termed ProbKT. The basic idea is formulating the weakly supervised detection problem as the probabilistic logical reasoning so that the detector can be trained with arbitrary types of weak supervision.  The output of the detector is fed into a neural probabilistic logical reasoning module. Several modifications are also proposed to reduce the computational cost of probabilistic programing specifically for object detection. Experiments show that ProbKT outperforms WSOD-transfer [41] and several baselines on CLEVR-mini, Molecules, MNIST on both detection and counting tasks.

**Summary Of The Review:**

Based on the novelty, the clarity and the thorough experiments, this paper is good for acceptance at ICLR.

---

> ### Author Response · Authors · 2022-11-15
> **Answers to reviewer's comments**
>
> We warmly thank reviewer q7Gr for their encouraging review ! Please find our response below
>
>
> ### About real-world evaluation
>
> We fully agree with the reviewer that the assessment of our approach on larger datasets with higher resolution images would be very interesting. We did not do so for two reasons. First, the motivation for our approach stems from the need to perform object detection in the low number of samples regime. The MNIST dataset contains only 700 images in the source and target domain, the Molecules dataset only 1400 images. Second, training objects detection models on large datasets like COCO requires a lot of compute (i.e. several days on a multiple GPUs) that we unfortunately cannot easily afford. We thus leave evaluation and inspection of our approach on larger datasets as an exciting future work direction.
> We also want to stress that the molecules recognition task is a real-world task from the chemoinformatics community. The (OOD) test images are real molecule images collected from the chemistry literature [1]. This further emphasizes the practical impact of our approach.
>
> [1] Noureddin M Sadawi, Alan P Sexton, and Volker Sorge. Chemical structure recognition: a rule-based approach. In Document Recognition and Retrieval XIX , volume 8297, page 82970E.International Society for Optics and Photonics, 2012.

---

### Official Review · Reviewer_Mz4p · 2022-10-25

**Confidence:** 4
**Correctness:** 4
**Technical Novelty And Significance:** 3
**Empirical Novelty And Significance:** 3
**Recommendation:** 6

**Clarity, Quality, Novelty And Reproducibility:**

The paper is well written and is novel in two points: the use of prior knowledge from other domains into object detection and learning in the second procedure is implemented in weakly supervised way.

**Strength And Weaknesses:**

The paper proposes a new framework that incorporates probabilistic logical reasoning method into the object detection models.
Strength:  Such a structure is easy to implement prior knowledge from other domains into deep learning models, is also able to train object detection models by leveraging richly annotated datasets from other domains and allowing arbitrary types of weak supervision on the target domain.
Weakness:
Although the proposed architecture integrates symbolic reasoning into deep learning architecture, it is still unclear from theoretical point of view that why the relabeling will provide performance improvement during symbolic reasoning.
With regard to "weakly supervises learning", the paper provides a simple example "The sum of all digitals in the image is considered as weak supervision". How does the cost function is implemented in the probabilistic logical reasoning stage
The proposed model includes the probabilistic reasoning procedure, which is of higher computational complexity.

**Summary Of The Paper:**

The paper proposes a framework based on probabilistic logical reasoning that allows to train object detection models with arbitrary types of weak supervision. The proposed architecture consists of two components: symbolic reasoning and deep learning architecture.
Such a structure is easy to implement weakly supervised learning in the target domain. Experimental results are given to demonstrate the good performance of the proposed model.
The main contribution of the paper is the incorporation of knowledge from other domains into object recognition framework. It is beneficial to use the proposed ProbKT to improve recognition performance on target domain and better generalization compared to existing baselines.



**Summary Of The Review:**

The use of probabilistic logical reasoning for improving the performance of object detection is interesting topic in the AI society.  The integration of probabilistic logical reasoning and deep learning model results in performance gain for object detection.

---

> ### Author Response · Authors · 2022-11-15
> **Answers to reviewer's comments**
>
> We thank reviewer Mz4p for their encouraging feedback that helped us improving the paper. Please find the answers to your comments below.
>
> ### About relabeling
>
> Indeed, the relabeling step is in theory not strictly necessary as one can backpropagate from the probabilistic logic head through the whole object detection backbone. Nevertheless, our experiments showed that backpropagating through the whole architecture is quite unstable. Instead, we saw that backpropagating through the classification heads was most effective. We then use relabeling as a way to update the whole object detection backbone.
>
> ### Elucidating the cost function in the probabilistic logic stage
>
> We added an example in Appendix E.1. For convenience, we state it also below.
> To illustrate the inference process let us follow the evaluation of the clause `sum([x1, x2],8)`, what can result from query `sum_digits(X,8)` in case of two visible digit in the image $X$.
>
> This clause is true if and only if $X_1 + X_2 = 8$.
>
> In case of MNIST digits ($\{0,1,\dots,9\}$) enumerating the possible worlds would give the following set:
>
> $\{(0,8), (1,7), (2,6), \dots, (8,0)\} $
>
> After summing the probability of all possible worlds we get:
>
> $ p_1(0)p_2(8) + p_1(1)p_2(7) + \dots + p_1(0)p_2(8),$
>
> where $p_1$ and $p_2$ are the distribution of random variable $X_1$ and $X_2$ respectively.
>
> Or in a general form:
>
>   $  p_Y(Y) =\sum_{X_1} p_1(X_1) p_2(Y - X_1).$
>
> As expected the distribution of the sum is the convolution of the distributions of the two terms. This observation trivially generalizes to more than two terms. The cost function corresponding to the maximum likelihood estimation is the negative log-likelihood $- \log(p_Y(Y))$.
>
>
> ### About the complexity of the probabilistic reasoning
>
> We acknowledge that the probabilistic reasoning step incurs additional computational complexity. Nevertheless, we show that this method is still practical, even in the case of many objects per image (e.g. the molecules dataset contains 15-20 molecules per image). What is more, we propose both a filtering approach that improves the computational complexity as well as a lightweight  variant of our method, ProbKT*.

---

> > ### Author Response · Authors · 2022-12-02
> > **small layout correction to our answer (part of text was shifted)**
> >
> > We would like to notify that we spotted a small layout issue in our answer above which we corrected. More specific, part of the text (elucidating the cost function in the probablistic logic stage) was shifted (to the end).

---

### Official Review · Reviewer_ijtT · 2022-10-26

**Confidence:** 3
**Clarity, Quality, Novelty And Reproducibility:** 1. The Novelty part is great; however…
**Correctness:** 3
**Technical Novelty And Significance:** 3
**Empirical Novelty And Significance:** 3
**Recommendation:** 6

**Details Of Ethics Concerns:**

Extensive content overlaps with the paper " "Updating Object Detection Models with Probabilistic Programming," even the write-up is copied from the paper.

**Strength And Weaknesses:**

Strength:
1. The introduced method for weakly supervised knowledge transfer learning is quite interesting.
2. The introduced method allows arbitrary types of supervision on the target domain can greatly alleviate the annotation burden.
3. Extensive experiments show the proposed method is quite effective.

Weakness:
1. The idea has been published before, and I do not see improvement or substantial differences over the published one.

2. The experiments are conducted on simpler content datasets, like MNIST. I expect some explanation of how to use the ProbKT for large-scale and complex content datasets on object detection, ex: what will the queries look like?
[ Authors provided feedback on this question, for instance, they said the model could be supervised on the count of detected objects, etc., on the COCO dataset. However, this counting information is already provided from the ordinary ground truth. Specifically, the number of bounding boxes for objects already provided the model with object counts. Therefore, when you supervised the model with queries like, “there are at least 5 dogs in this image” or “there are more humans than animals in this image,” what additional information did we provide to the model? In other words, I am still not fully convinced of the author's answer.]


**Summary Of The Paper:**

Detailed annotations are often not available in practice, and only high-level image information is available on the target dataset; this work presents a fine-tuning method using ProbKT to update object detection models from a pre-trained architecture. Nevertheless, the presented knowledge transfer method can be built upon arbitrary types of weak supervision on the target domain, such as complex logic statements. The authors empirically show on three types of datasets that fine-tuning with ProbKT leads to significant improvement in the target domain and gives better generalization compared to existing baselines.

**Summary Of The Review:**

Overall, the presented idea is novel and helps equip the model with logical reasoning as humans on arbitrary input. The experiments show its effectiveness, especially I can imagine it can help more when the quantity of the dataset is not large enough to train a robust model.

---

> ### Author Response · Authors · 2022-11-15
> **Answers to reviewer's comments**
>
> We first want to thank reviewer ijtT for their insightful review. Please find our answers below
>
> ### About large-scale and complex datasets.
>
> The queries do not depend on the scale of the dataset but rather on the available supervision signal. For example, in the COCO dataset, we could accommodate queries/weak labels such as “there are at least 5 dogs in this image” or “there are more humans than animals in this image”.
>
> We want to stress the difference between the complexity of the images and the complexity of the supervision signal. While MNIST digits are examples of simple images, the supervision that we used in this example (only the sum of the digits is available) is arguably complex. In fact, we show that no other competitor method is able to handle such a supervision signal. Our method can be implemented over any existing object detection model, therefore can handle complex images as long as there exists a model architecture that can detect objects on them.
> What is more, we want to stress that the molecule recognition task is a real-world task from the chemoinformatics community. The (OOD) test images are real molecule images collected from the chemistry literature [1].
>
> Regarding larger-scale dataset, we want to clarify that the queries are not dependent on the number of images but rather on the type of supervision that is available for object detection. As such, arbitrarily complex supervision can also be demonstrated on very simple images (e.g. the MNIST dataset discussed above).
>
> Lastly, the computational complexity of our probabilistic logic part and the object detection are additive. That means that training our method on large scale datasets does not imply any additional difficulty, compared to other state of the art object detection architectures. That said, training classical object detection models is notoriously computationally heavy (i.e. several days of GPU for each model). Instead, we focused our efforts on small scale, yet real-world datasets.
>
> [1] Noureddin M Sadawi, Alan P Sexton, and Volker Sorge. Chemical structure recognition: a rule-based approach. In Document Recognition and Retrieval XIX , volume 8297, page 82970E.International Society for Optics and Photonics, 2012.
>
> ### About code sharing
>
> As written in our abstract, we released the code in the submission in an anonymous repo : https://github.com/iclr2023-ProbKT/ProbKT

---

> > ### Comment · Reviewer_ijtT · 2022-11-22
> > **Receive the author feedback and modify my rating**
> >
> > Thanks for the clarification on my questions. Please consider citing your previous workshop paper in the reference if this submission is accepted.

---

> ### Author Response · Authors · 2022-12-07
> **Additional clarification to reviewer ijtT**
>
> Dear reviewer,
>
> We just spotted that there was still an open question in your review.
>
> `However, this counting information is already provided from the ordinary ground truth. Specifically, the number of bounding boxes for objects already provided the model with object counts. Therefore, when you supervised the model with queries like, "there are at least 5 dogs in this image" or "there are more humans than animals in this image," what additional information did we provide to the model? In other words, I am still not fully convinced of the author's answer.`
>
> We apologize if our previous answer may have been confusing. In the case of COCO data set more detailed labels are indeed already available.  However our point is to illustrate how a model already pre-trained on COCO data set could be transferred to another domain where only weakly labeled samples are available (e.g. “there are at least 5 dogs in this image” or “there are more humans than animals in this image”). Our work focuses on the domain adaptation task where models trained on richly annotated datasets like COCO can be adapted to a new domain where less detailed supervision is available.
>
> Regarding the workshop paper, we will add it to our references.
>
> Please let us know if any questions of concerns remain regarding our work.
>
> Best Regards,
>
> The Authors

---

### Author Response · Authors · 2022-11-18
**Summary of answers**

Dear reviewers and area chair,

We want to thank you all again for your time and effort in reviewing our work. As the rebuttal period comes to a close we would like to summarize our responses to the comments you raised in your reviews:

- We have provided more details about the probabilistic logical programming engine and the inference process. We have now added a detailed illustration of the inference process in Appendix E. We have also clarified our dependence on DeepProbLog in Section 3. While our implementation currently uses DeepProbLog for the underlying probabilistic logical reasoning step, our architecture can be envisioned with a different choice of engine.

- We have highlighted the motivation behind weak supervision in objects detection tasks and the pertinence of our method for real-world application. Notably, the chemical entity recognition use case (Molecules Dataset) is an important real-world task which is getting increasing attention in the community. See for instance [1], a forthcoming NeurIPS paper.

- We have clarified the expressivity of our method in Section 2 of the paper. Our model inherits the expressivity of Problog and is thus based on a subset of first-order predicate logic.

- We have improved the overall clarity of the paper. In particular, we have clarified the exact query used in the illustrative Figures 1, 5 and 6. We have also repositioned the related works section.

We are available for elucidating any other concern or question you might have.

Best Regards,

The authors

[1] CEDe: A collection of expert-curated datasets with atom-level entity annotations for Optical Chemical Structure Recognition.” Thirty-sixth Conference on Neural Information Processing Systems Datasets and Benchmarks Track, 2022.

---

### Decision · Program_Chairs · 2023-01-20

**Decision:**

Accept: poster

**Justification For Why Not Higher Score:**

The paper presents an application of existing techniques to a new problem. Thus novelty is limited.

**Justification For Why Not Lower Score:**

The application of neuro-symbolic methods to domain adaptation is well done and interesting. Results are good. All reviewers agree that the paper should be accepted.

**Metareview: Summary, Strengths And Weaknesses:**

The paper presents a well done application of deep Problog to the problem of domain adaptation. The authors propose adapting pre-trained models using weak feedback in the form of complex logical statements.

All reviewers agree that the paper should be accepted. While the main technique is not novel (it is essentially deep problog), the authors show how it can be successfully applied to domain adaptation problems, and achieve good performance on a number of problems.

**Note From Pc:**

if the above contains the word "oral" or "spotlight" please see: "oral" presentation means -> notable-top-5% and "spotlight" means -> notable-top-25%. As stated in our emails, we are disassociating presentation type from AC recommendations